# Dislodgment Resistance, Adhesive Pattern, and Dentinal Tubule Penetration of a Novel Experimental Algin Biopolymer-Incorporated Bioceramic-Based Root Canal Sealer

**DOI:** 10.3390/polym15051317

**Published:** 2023-03-06

**Authors:** Galvin Sim Siang Lin, Norhayati Luddin, Huwaina Abd Ghani, Josephine Chang Hui Lai, Tahir Yusuf Noorani

**Affiliations:** 1Department of Dental Materials, Faculty of Dentistry, Asian Institute of Medicine, Science and Technology (AIMST) University, Bedong 08100, Malaysia; 2Prosthodontics Unit, School of Dental Sciences, Universiti Sains Malaysia, Health Campus, Kubang Kerian, Kota Bharu 16150, Malaysia; 3Conservative Dentistry Unit, School of Dental Sciences, Universiti Sains Malaysia, Health Campus, Kubang Kerian, Kota Bharu 16150, Malaysia; 4Department of Chemical Engineering and Sustainable Energy, Faculty of Engineering, Universiti Malaysia Sarawak, Kota Samarahan 94300, Malaysia

**Keywords:** alginate, biomaterials, biopolymer, bond strength, dentistry, endodontics, hydrogel

## Abstract

The currently available bioceramic-based sealers still demonstrate low bond strength with a poor seal in root canal despite desirable biological properties. Hence, the present study aimed to determine the dislodgment resistance, adhesive pattern, and dentinal tubule penetration of a novel experimental algin-incorporated bioactive glass 58S calcium silicate-based (Bio-G) sealer and compared it with commercialised bioceramic-based sealers. A total of 112 lower premolars were instrumented to size 30. Four groups (*n* = 16) were assigned for the dislodgment resistance test: control, gutta-percha + Bio-G, gutta-percha + BioRoot RCS, and gutta-percha + iRoot SP, with exclusion of the control group in adhesive pattern and dentinal tubule penetration tests. Obturation was done, and teeth were placed in an incubator to allow sealer setting. For the dentinal tubule penetration test, sealers were mixed with 0.1% of rhodamine B dye. Subsequently, teeth were cut into a 1 mm-thick cross section at 5 mm and 10 mm levels from the root apex, respectively. Push-out bond strength, adhesive pattern, and dentinal tubule penetration tests were performed. Bio-G showed the highest mean push-out bond strength (*p* < 0.05), while iRoot SP showed the greatest sealer penetration (*p* < 0.05). Bio-G demonstrated more favourable adhesive patterns. No significant association was noted between dislodgment resistance and dentinal tubule penetration (*p* > 0.05).

## 1. Introduction

A proper three-dimensional seal of the prepared root canal system is necessary for successful endodontic treatment to prevent bacterial reinvasion via microleakage [1]. Although gutta-percha is still the most commonly used core filling material in such a treatment, root canal sealers are needed, as the lack of bonding and adhesion between gutta-percha and the root dentinal walls poses a challenge in forming a hermetic seal in the root canal system [2]. Hence, a root canal sealer is employed in this situation to provide a fluid-tight seal at the gutta-percha core-sealer and dentin-sealer interfaces [3]. It is worth noting that a fluid-tight seal is crucial for enhancing the sealer materials’ bond strength and preventing dislodgment under high occlusal stresses [1]. In addition, root canal sealers fill the anatomical irregularities within the complex root canal system, provide a certain degree of dentinal tubule penetration, and minimise microleakage [4].

A variety of root canal sealers have been introduced into the market, including epoxy resin-based, methacrylate resin-based, calcium hydroxide-based, glass ionomer cement-based, silicon-based, zinc oxide eugenol-based, and bioceramic-based (which includes mineral trioxide aggregate and pure calcium silicate) sealers [5]. Bioceramic-based root canal sealers have gained popularity among clinicians, and a systematic review has shown that root canal obturated with bioceramic-based sealer was associated with significantly lower short-term postoperative pain accompanied with lower analgesic intake and flare-up incidence, as compared to root canal-treated teeth obturated with other sealers [6]. Bioceramic materials based on mineral trioxide aggregate (MTA) were initially developed at Loma Linda University in the 1990s, and later they were modified into root canal sealers to offset the shortcomings of resin-based sealers [7]. Nevertheless, MTA exhibits poor flowability with the presence of heavy metals, leading to the development of new-generation bioceramic sealers based on pure calcium silicate cement [8,9].

iRoot SP (BioCeramix Inc. in Vancouver, BC, Canada) is an injectable calcium silicate-based root canal sealer consisting of calcium silicate, calcium phosphate, calcium hydroxide, zirconium oxide as a radiopacifier, filler, and thickening agents [9]. Unlike other calcium silicate sealers, IRoot SP contains monobasic calcium phosphate, which has been claimed to facilitate the reaction with calcium hydroxide and form hydroxyapatite during its hydration [10]. In 2016, a new root canal sealer based on tricalcium silicate, BioRoot RCS (Septodont, Saint-Maur-des-Fossés Cedex, Paris, France), was released [11]. The liquid portion consists of calcium chloride and polycarboxylate, while the powder is made up of tricalcium silicate, calcium phosphate, povidone, and zirconium dioxide [3]. It has been reported that BioRoot RCS displayed exceptional push-out bond and excellent sealing ability, making it an excellent bioceramic sealer of choice [1]. Furthermore, BioRoot RCS has a lower concentration of heavy metals such as lead, chromium, and arsenic as compared to other MTA products [12]. This is crucial for achieving a highly predictable outcome when employing biomaterials for root canal treatment. In 2022, a new experimental bioceramic-based sealer was invented by incorporating alginate biopolymer into bioactive glass 58S and calcium silicate [13].

Algin, also known as alginic acid, is a hydrophilic polymer derived from seaweed that forms a viscous gel-like structure when hydrated [14,15]. When combined with sodium or calcium, it can undergo gelation and produce salts known as alginates. Alginate gelation occurs when divalent cations, such as calcium ion, Ca^2+^, bind to alginate and create an insoluble diamond-shaped hole with a hydrophilic cavity that binds the Ca^2+^ by multicoordinating the oxygen atoms from the carboxyl groups [14]. Furthermore, alginate is a nontoxic, commonly accessible, biocompatible, and nonimmunogenic marine biopolymer [16]. Alginates have several free hydroxyl and carboxyl groups scattered throughout their backbone, making them very reactive and prone to strong cross-linking with other particles. Alginate is frequently employed in medical applications including wound healing, medication administration, and tissue engineering due to its unique features [17]. Moreover, regardless of temperature, algin can form strong intermolecular cross-linking with a shorter setting time, which is of primary interest in current clinical practise, as this will further enhance the intermolecular cohesiveness, preventing material dislodgment from the root canal walls and making it operator-friendly due to its fast setting [14]. However, the use of alginate hydrogel in endodontics, particularly for root canal sealers, is still considered new, and there is little evidence available in the literature. Therefore, it is possible to speculate that this compact gel-like framework will enable adequate root canal system sealing.

Push-out bond strength tests are typically used to assess resistance of sealers being forced out of the root dentine wall [1,3,9]. These are mechanical tests in which the gutta-percha and sealer are pushed out or dislodged by applying a longitudinal tensile load to the long axis of the root sample. As fracturing occurs parallel to the dentine-bonding interface and the push-out bond strength test is repeatable, it can be used to evaluate parallel-sided samples even when the bond strength is low [18]. Another significant consideration when assessing root canal sealers is their capacity to penetrate the dentinal tubule and create a solid physical barrier, which enhances the retention of the root filling materials and entombs remaining microorganisms in the root canal [2]. Additionally, given that the sealers may penetrate deep inside the tubules, it is reasonable to assume that their antibacterial impact will function better if present [19].

An ideal root canal sealer should not only offer outstanding biocompatibility; it also needs to have excellent dentinal tubule penetration, adhesiveness, and bonding with the root canal walls to prevent dislodgment through gap formation at the sealer-wall interface. Therefore, the purpose of the present in vitro study was to determine the dislodgment resistance and dentinal tubule penetration of a novel experimental algin biopolymer-incorporated bioactive glass 58S calcium silicate-based (Bio-G) sealer with other commercially available bioceramic-based sealers, namely BioRoot RCS and iRoot SP. The first null hypothesis was that there is no significant difference between novel experimental Bio-G and two other commercialised bioceramic-based sealers in terms of dislodgment resistance to the root dentinal wall. The second null hypothesis was that there is no significant difference between novel experimental Bio-G with other commercialised bioceramic-based sealers in terms of adhesive pattern. The third null hypothesis was that there is no significant difference between novel experimental Bio-G with other commercialised bioceramic-based sealers in terms of dentinal tubule penetration.

## 2. Materials and Methods

### 2.1. Sample Size Calculation

Based on a previous similar study [1], the sample size was determined using the Bivariate Normal Distribution Model and Correlation (G*Power 3.1.9.7 for Windows, Heinrich-Heine-Universität, Düsseldorf, Germany), with values of effect size set at 0.456, *α* = 0.05, *β* = 0.85, and correlation P for H_0_ = 0 [20]. Taking into consideration the additional 15% for potential dropout, these data were entered into an F-test family utilising a priori power analysis. The sample size generated for the dislodgment resistance test was 64 teeth. Meanwhile, the same method was used to generate the sample size for the dentinal tubule penetration test using the information obtained from another previous study with the effect size of 0.454, *α* = 0.05, *β* = 0.85, and correlation P for H_0_ = 0 [4]. The sample size for dentinal tubule penetration with 15% dropout was set at 48 teeth. Hence, the total sample size for the current study was 112 teeth.

### 2.2. Sample Preparation

The present in vitro experimental study was approved by the Human Research Ethics Committee Universiti Sains Malaysia on 22 May 2021 (Reference No.: USM/JEPeM/21060495). The flowchart of the study is illustrated in Figure 1. Freshly extracted mandibular premolars were collected from patients between the ages of 20 and 40 who attended the university’s dental clinic for tooth extraction due to orthodontic or periodontal reasons. All collected teeth were examined by a single-blinded examiner using a microscope (Leica DM 300, Leica Microsystem GmbH, Wetzlar, Germany) at a magnification of 20× to screen for root caries, fractures, abrasions, and restorations [21]. Next, the teeth were measured to ensure that the total tooth length was 21 mm (±1 mm) and the root length was 12 mm (±1 mm). Subsequently, the teeth were radiographically inspected (X-ray Unit, Planmeca, Helsinki, Finland) to confirm the existence of a single Vertucci’s Type I canal with mature apical foramen. All teeth were then immersed in 2.5% sodium hypochlorite (NaOCl, Lenntech, Delfgauw, The Netherlands) for 24 h to remove remaining tissue debris [1].

Consequently, access cavity was performed for each tooth using an endo access bur (Size 4, 21 mm, Dentsply Maillefer, Ballaigues, Switzerland), and the canal patency was examined with a size 10 K-file (FlexOFiles; Dentsply Maillefer, Ballaigues, Switzerland). The working length was standardised at the apical foramen, and a crown-down technique was used to shape all the canals with NiTi rotary files up to size 30, taper 0.04 (T-Flex, Shenzhen Perfect Medical Instruments Co., Ltd., Shenzhen, China). An amount of 5 mL of a 17% ethylenediaminetetraacetic acid (EDTA) solution (Pulpdent, Watertown, MA, USA) was employed as the chelating agent to remove smear layers after copious irrigation with 2.5% NaOCl solution during each file instrumentation. The remaining EDTA in the canals was flushed out using 10 mL of normal saline solution (EYE-SNS120, Promed Marketing Sdn Bhd, Subang Jaya, Malaysia) as the final irrigant [21], and all canals were dried with size 30 paper points (Dentsply, Maillefer, Ballaigues, Switzerland).

### 2.3. Dislodgment Resistance Test

A total of 64 mandibular premolars were randomly sorted into 4 groups of 16 tooth samples each. They were categorised as:

Group 1—gutta-percha only (control);

Group 2—gutta-percha + experimental bioceramic-based sealer (Bio-G);

Group 3—gutta-percha + BioRoot RCS;

Group 4—gutta-percha + iRoot SP.

Prior to obturation, the master gutta-percha size 30 tapered 0.04 was used to examine the presence of ‘tug-back’ [1], and sealers were prepared in accordance with the manufacturer’s instructions. Bio-G was mixed based on the techniques proposed by a previous study [13], whereas the premixed iRoot SP was injected directly into the canals. A single-cone obturation technique was used, and the sealer materials were first applied around the canal walls using matched-taper gutta-percha cones before obturation. Following that, the access cavities were totally etched with 37% phosphoric acid (Gel Etchant, Kerr Corporation, Orange, CA, USA) for 15 s before being washed, dried, and coated with bonding agent (OptiBond™ Universal, Kerr Corporation, Orange, CA, USA). Then, the cavities were restored with nanohybrid resin composites (Filtek Z250 XT, 3M ESPE, Saint Paul, MN, USA) incrementally and light-cured for 40 s. The final restorations were polished using composite polishing kits (PN 0310BB, Shofu, San Marcos, CA, USA). To minimise internal bias, all experimental procedures were carried out by a single expert operator. All tooth samples were placed in an incubator (Memmert GmbH + Co. KG, Schwabach, Bavaria, Germany) for 72 h at 37 °C and 95% humidity to allow the setting of sealer materials [1].

After 72 h, a hard tissue cutter (EXAKT 312, EXAKT Technologies, Inc., Oklahoma City, OK, USA) was used to cut the tooth samples into a 1 mm-thick cross section at 5 mm and 10 mm from the root apex, which represents the middle and coronal third root regions, respectively. Only the coronal and middle third root sections were chosen for the present study, as the amount of sealer materials in the apical third regions was too minimal to be evaluated [22]. The sample surfaces were then polished with increasing grit sandpaper (P600 to P2000; 3M ESPE, Saint Paul, MN, USA). Prior to push-out testing, the samples were viewed under the same Leica microscope to confirm that the obturating materials were free of voids and cracks. Samples were then subjected to increasing occlusal push-out force using a spherical steel point of 0.6 mm and 0.4 mm diameter for the coronal and middle third sections, respectively. The push-out test was carried out using a Universal Testing Machine (AGX-V series, Shimadzu, Kyoto, Japan) with a speed setting of 1 mm/min until the gutta-percha and sealer were pushed out. Due to the tapering nature of the root canals, the push-out force was applied from apical to coronal direction. The maximum force required to cause the gutta-percha and sealer materials to be dislodged was measured in Newtons (N).

Push-out strength (MPa) was calculated as maximum loading force (N)/dentine wall surface area (mm^2^) [1]. A conical frustum surface area formula was used to determine the surface area (mm^2^):(1)$$\pi \;\times \;\left(r_{1}+r_{2}\right)\times \sqrt{{\left(r_{1}-r_{2}\right)}^{2}+h^{2}}$$
whereby *r*_1_ is the radius at the coronal part, *r*_2_ is the radius at the apical part, and *h* is the thickness of the sample, which is 1 mm.

### 2.4. Adhesive Pattern Test

The remaining sealer adhering to the root dentine walls was examined under a Leica microscope at 20× magnification using a simple classification as proposed in a previous study [1]. The sealer adhesion to root dentinal walls was divided into four quadrants, with Type 1 as the least favourable and Type 4 as the most favourable. ‘Non-adhesive’ was classified if there was no sealer noted on the root dentine wall.

### 2.5. Dentinal Tubule Penetration Test

The dentinal tubule penetration was evaluated using a confocal laser scanning microscopy (Leica TCS SP5 II, Leica Microsystem GmbH, Wetzlar, Germany). Another 48 mandibular premolars were randomly assigned into 3 groups, each consisting of 16 teeth. The groups were:

Group A—gutta-percha + experimental bioceramic-based sealer (Bio-G);

Group B—gutta-percha + BioRoot RCS;

Group C—gutta-percha + iRoot SP.

The sealer materials were mixed according to each manufacturer’s instructions, with 0.1% of rhodamine B dye (C.I.45170, Sigma-Aldrich, Merck KGaA, Darmstadt, Hesse, Germany) added to provide fluorescence [2]. Obturation was performed using a single-cone technique with matched-taper gutta-percha cones as per the descriptions in the dislodgment resistance test. The access cavities were then restored using the same nanohybrid resin composites, light-cured, and polished before placing them in the incubator for 72 h at 37 °C and 100% humidity.

After 72 h, the tooth samples were cut into 1 mm-thick cross sections at 5 mm and 10 mm from the root apex. The specimen surfaces were then polished using sandpaper prior to being viewed under the confocal scanning electron microscopy at 10× magnification. For the rhodamine B dye, 557 nm and 577 nm were chosen as the excitation and emission wavelengths, respectively [7]. Images were recorded at fluorescent mode and a numeric aperture of 0.3 and 1.3 mm, respectively [23]. The maximum depth of sealer penetration was measured from the canal wall to the deepest point, while the mean sealer penetration depths were measured by averaging the penetration depths at 4 circumferential points, 12, 3, 6, and 9 o’clock, corresponding to the buccal, mesial, lingual, and distal directions, respectively [2].

### 2.6. Statistical Analysis

Data analyses were carried out using SPSS version 26.0 for Windows (SPSS Inc., Chicago, IL, USA). The Shapiro–Wilk test was used to verify the normality of the data distribution. Since the *p* values for the control, Bio-G, BioRoot RCS, and iRoot SP were 0.301, 0.604, 0.102, and 0.06, respectively, the normality null hypothesis was accepted, suggesting that a normal distribution was observed. The dislodgment resistance and dentinal tubule penetration were analysed using one-way ANOVA in conjunction with a post hoc Tukey HSD test, since the data were normally distributed. Meanwhile, the adhesive pattern was evaluated using a chi-square test. A Pearson correlation coefficient (*r*) test was performed to identify the correlation between sealer’s dislodgment resistance and tubular penetration. The significance level selected was *p* = 0.05.

## 3. Results

### 3.1. Dislodgment Resistance

The dislodgment resistance results are presented in Table 1 and depicted in Figure 2. A significant difference (*p* < 0.001) was noted among the push-out bond strength of all sealer groups at the 10 mm level, with Bio-G showing the highest mean push-out bond strength, followed by BioRoot RCS, iRoot SP, and finally the control group. However, multiple comparisons showed that no significant difference was found between Bio-G and BioRoot RCS (*p* = 0.238) or between BioRoot and iRoot SP (*p* = 0.152). Similar patterns were noted at the 5 mm level whereby Bio-G exhibited greater mean push-out bond strength, followed by BioRoot RCS, iRoot SP, and the control group, but no significant difference was noted between Bio-G and BioRoot RCS (*p* = 0.999) or between BioRoot RCS and iRoot SP (*p* = 0.061).

### 3.2. Adhesive Pattern

The adhesive pattern of each sealer group is demonstrated in Table 2 and illustrated in Figure 3. At the 5 mm level, most of the samples in Bio-G and BioRoot RCS fall into Type 3 and Type 4 adhesive patterns, with only two samples in BioRoot RCS showing Type 2 adhesive patterns. However, iRoot SP showed a greater frequency of Type 1 and Type 2 adhesive patterns. At the 10 mm level a similar pattern was noted, with most of the samples in Bio-G and BioRoot RCS falling in Type 3 and Type 4 adhesive patterns and only three samples in BioRoot RCS and one sample in Bio-G showing Type 2 adhesive patterns. Nonetheless, iRoot SP showed a greater degree of Type 1 and Type 2 adhesive patterns. Overall, Bio-G exhibited significantly (*p* < 0.001) better adhesive properties than BioRoot RCS and iRoot SP at both the 5 mm and 10 mm levels from the root apex.

### 3.3. Dentinal Tubule Penetration

The maximum and mean tubular penetration (Figure 1) depths at the 5 mm and 10 mm levels are listed in Table 3. Representative tubular penetration images are illustrated in Figure 4. Statistically significant differences were noted among all sealer groups (*p* = 0.023), with iRoot SP showing the greatest sealer penetration, followed by Bio-G and BioRoot RCS at the 5 mm level. Similarly, iRoot SP demonstrated the highest mean sealer penetration at the 5 mm level (*p* = 0.001) from the root apex, followed by Bio-G and, finally, BioRoot RCS. However, no statistically significant difference was noted between Bio-G and BioRoot RCS (*p* = 0.687) at the 5 mm level.

At the 10 mm level, a statistically significant difference was noted among all sealer groups (*p* = 0.001), with iRoot SP showing the greatest sealer penetration, followed by Bio-G and BioRoot RCS, but no significant difference was found between Bio-G and iRoot SP (*p* = 0.663). Similarly, iRoot SP demonstrated the highest mean sealer penetration at the 10 mm level (*p* = 0.043) from the root apex, followed by Bio-G and, finally, BioRoot RCS. The Pearson’s correlation coefficient suggested that no significant association was noted between the dislodgment resistance and dentinal tubule penetration of all sealer materials at the 10 mm level (*r* = 0.187, *p* = 0.203) and the 5 mm level (*r* = 0.131, *p* = 0.373) from the root apex, respectively.

## 4. Discussion

The present study evaluated and compared the dislodgment resistance, adhesive pattern, and dentinal tubule penetration of novel experimental algin-incorporated bioactive glass 58S calcium silicate-based sealer with two other commercially available bioceramic-based root canal sealers. The first null hypothesis was partially rejected, as Bio-G sealer demonstrated significantly higher push-out bond strength than iRoot SP and the control group. A plausible explanation would be the presence of algin biopolymer in Bio-G sealer, a hydrophilic polysaccharide that creates a viscous gel-like structure when hydrated [13]. The free hydroxyl and carboxyl groups in hydrocolloid algin are believed to react actively and form strong intermolecular cross-linkage with the calcium silicate particles, known as the calcium silicate–alginate hydrogel polymer. The tightly bound configuration between the alginic acid and calcium ions can result in a compact gel network [14]. Hence, one may postulate that adequate dislodgment resistance of the sealer to root dentinal walls can be accomplished with this compact gel-like framework. Instead of shrinking, algin displays slight expansion owing to its hydrophilic functional groups [15,17], and it can be anticipated that this slight expansion can compensate for the voids that exist in the root canal system, thus offering a better seal.

Although iRoot SP in the current study was found to exhibit the lowest push-out bond strength, previous studies have shown that iRoot SP performed well in terms of dislodgment resistance as compared to resin-based and calcium hydroxide-based sealers [10,24]. Generally, bioceramic-based sealers can form mineral plugs at the root dentinal wall interface due to the formation of apatite layers, intrafibrillar apatite deposition, and tag-like structures, referred as the mineral infiltration zone [1,25]. This apatite nucleation prevents sealer dislodgment from the dentinal walls and gutta-percha, which can create a tight seal with minimal shrinkage in the root canal system [10,11]. Even though bioceramic-based sealer materials can form colloidal calcium silicate hydrate gels that harden and promote micromechanical interaction via tag-like structures [3], the explanation for the dislodgment resistance of bioceramic sealers based on their hydraulic setting reaction is dubious, and the exact mechanism for bonding such sealers to root dentine walls is still unknown [1].

The push-out test is still a valuable technique for comparing and ranking the bonding capabilities of various endodontic sealers. It has been argued that the presence of gutta-percha in push-out bond strength tests is not recommended and that completely filled root canals with sealer is advocated, but simulating the clinical settings of a real root canal in such a situation is no longer possible [26]. Thus, gutta-percha was used in the current study to mimic the actual clinical conditions in a root canal treatment. The present study used punch diameters of 0.6 mm and 0.4 mm for the samples at the 10 mm and 5 mm levels from the root apex, respectively. These diameter values are within the permissible range, because it has been proposed that the results will not be significantly affected by a punch diameter between 70% and 90% of the canal diameter [27]. Additionally, a 0.04 taper of a NiTi file was utilised to allow for a negligibly small impact on frictional resistance during the push-out test [1,3,27]. Moreover, the current study employed a single cone technique during root canal obturation, as both warm vertical and lateral condensation were found to have an adverse effect on the push-out testing and are less reproducible in push-out testing [3]. Another important parameter that also needs to be taken into consideration when performing dislodgment resistance studies on different root canal sealers is the use of an irrigating solution, as various irrigating solutions may have an impact on the adhesion of sealer materials to root dentinal walls [28].

The second null hypothesis was also partially rejected. Bio-G sealer showed the most favourable adhesive pattern (Type 3 and Type 4). One explanation could be due to the formation of calcium–silicate–alginate during the setting reaction that resulted in an increase of the biopolymer swelling capacity [29]. This emphasises the significance of incorporating algin into sealer materials to increase their bond strength and adhesiveness to the dentinal walls of root canals. iRoot SP, on the other hand, demonstrated the highest frequency of unfavourable adhesive patterns (Type 1 and Type 2), which corroborates with its bond strength values. Accordingly, it seems reasonable to argue that a more favourable adhesive pattern exists when the sealer materials display superior dislodgment resistance to root dentinal walls. Adhesion is influenced by several variables, including the adherend’s surface energy (gutta-percha or root dentine), the adhesive’s surface tension (root canal sealer), the adhesive’s capacity to moisten the surfaces, and the cleanliness of the adherend surface [30,31]. Thus, one can conclude that the intricate process of sealers adhering to root dentinal walls and gutta-percha involves the use of dentine pretreatment techniques, physical characteristics, and chemical sealer components.

The third null hypothesis was partially rejected. In the present study, Bio-G generally showed greater sealer penetration than BioRoot RCS, which could be due to the particle size distribution, as Bio-G was found to have a smaller particle size than BioRoot RCS [13]. Furthermore, BioRoot RCS was reported to feature a greater film thickness, which hindered its ability to flow and penetrate the dentinal tubules [19,32]. Due to insufficient details on the novel experimental sealer, a direct comparison with outcomes from other studies is not feasible. On the other hand, iRoot SP exhibited higher mean tubular penetration depths due to its high flowability, which is consistent with other similar studies [33,34]. Another explanation could be due to the fact that iRoot SP is a premixed bioceramic-based sealer as opposed to Bio-G and BioRoot RCS, and hence the sealer’s consistency and viscosity during mixing may be a challenge for optimal standardisation. Nevertheless, it can be asserted that the current findings support the notion that smaller particle size and increased fluidity are essential characteristics in bioceramic-based sealers that allow them to create more sealer tags when in contact with the root dentinal walls, leading to better sealer penetration and adaptation [2]. Greater dislodgment resistance and tubular penetration are made possible by the discovery of bioceramic-based sealers that exhibit strong hydraulic conductivity, which can form a tag-like mineral infiltration zone and clog the dentinal tubules [1].

The current findings show that both the maximum and mean dentinal tubule penetrations were higher at the 10 mm level than the 5 mm level due to the decreased number of dentinal tubules apically, which is consistent with a previous study [2]. A disparity of the sealer penetration depths can also be influenced by the ‘butterfly effect’ present in the root dentine. The average butterfly effect in lower premolars was found to be approximately 40%, with the coronal root regions showing greater tubular density [21]. Thus, it is conceivable to hypothesise that a greater sealer penetration would occur in root regions with higher tubular density and that a deeper sealer penetration would be observed in the buccolingual direction. All tooth samples in the current study were irrigated with 10 mL of normal saline solution as the final irrigating solution to flush out remaining EDTA and eliminate intrinsic calcium ions that may impact the sealer’s ability to penetrate the dentinal tubules [4]. Other factors such as the physicochemical properties of the sealer, the presence of a smear layer, the anatomical configuration of the root canal, and the obturation techniques may also affect dentinal tubule penetration [7,35]. Nevertheless, the current study revealed no correlation between the dislodgment resistance and dentinal tubule penetration of root canal sealers, which corroborates a previous similar study [35]. It is also crucial to mention that sealer penetration depth may not correlate with its ability to adapt and seal the root canal system [4].

The tubular penetration of sealer materials to root dental walls has been determined using a variety of microscopic methods, such as scanning electron microscopy (SEM) and CLSM [36,37]. Although a recent review found that SEM is more sensitive in detecting tubular penetration and providing detailed information due its higher level of magnification [38], CLSM was chosen as the evaluation method in the current study. This is because SEM exhibits a number of flaws, such as lacking precise identification at lower magnifications and the production of artefacts when preparing tooth samples for analysis [39]. Furthermore, the lack of fluorescent markers makes it difficult to distinguish among sealer materials that are present in the root dentinal walls, and inexperienced researchers may have difficulty interpreting SEM images. In contrast, CLSM eliminated the demand for sample specimen processing, such as removing the smear layer, and allowed optical regions beneath the surface of dentine to be seen [4,33]. This is crucial, as removing the smear layer could potentially cause the calcium silicate-based sealer on the sample’s surface to be eradicated. Moreover, CLSM does not require the tooth samples to be dehydrated and minimises the risk of bioceramic-based sealer material deformation [38]. Due to the significant contrast of the dye used, CLSM also provides a full representation of interfacial adaptation and sealer dispersion, enabling adequate analysis [40]. Owing to these advantages, CLSM was used in several in vitro experimental studies [41,42,43]. Nonetheless, a standardised evaluation method that can assess three-dimensional dentinal tubular penetration is necessary in future research.

Several limitations could be found in the present study. The findings of the current study may not be appropriate for being fully translated to a clinical situation due to the lack of simulated periodontal ligament. Moreover, the sealer adaptation at the gutta-percha and root dentinal wall interfaces was not examined, rendering future exploration because the presence of gaps in root canal filling material may promote the growth of microorganisms and jeopardise the success of the root canal treatment [2]. Nonetheless, the present study offers insights and reproducible results that can be applied to compare different root canal sealers and establish standard benchmarks for future investigation on bioceramic-based sealers. More studies are warranted to evaluate the physical, mechanical, chemical, and biological properties of the novel experimental sealer as well as its long-term sealer adhesion, sealing ability, and clinical performance.

## 5. Conclusions

Within the limitations of the present study, the novel Bio-G sealer showed more favourable adhesive pattern, with comparable dislodgment resistance to BioRoot RCS and significantly greater bond strength, than iRoot SP. Although iRoot SP demonstrated greater tubular penetration, Bio-G and iRoot SP both exhibited similar mean dentinal tubule penetration in the coronal third root region. Moreover, the dislodgment resistance of bioceramic-based sealers is independent of their ability to penetrate dentinal tubules. In short, the acceptable dentinal tubule penetration of Bio-G, along with its outstanding dislodgment resistance, may aid in strengthening the seal of the root canal system. However, such a conclusion needs further verification with more well-designed and well-controlled clinical research.

## Figures and Tables

**Figure 1 polymers-15-01317-f001:**
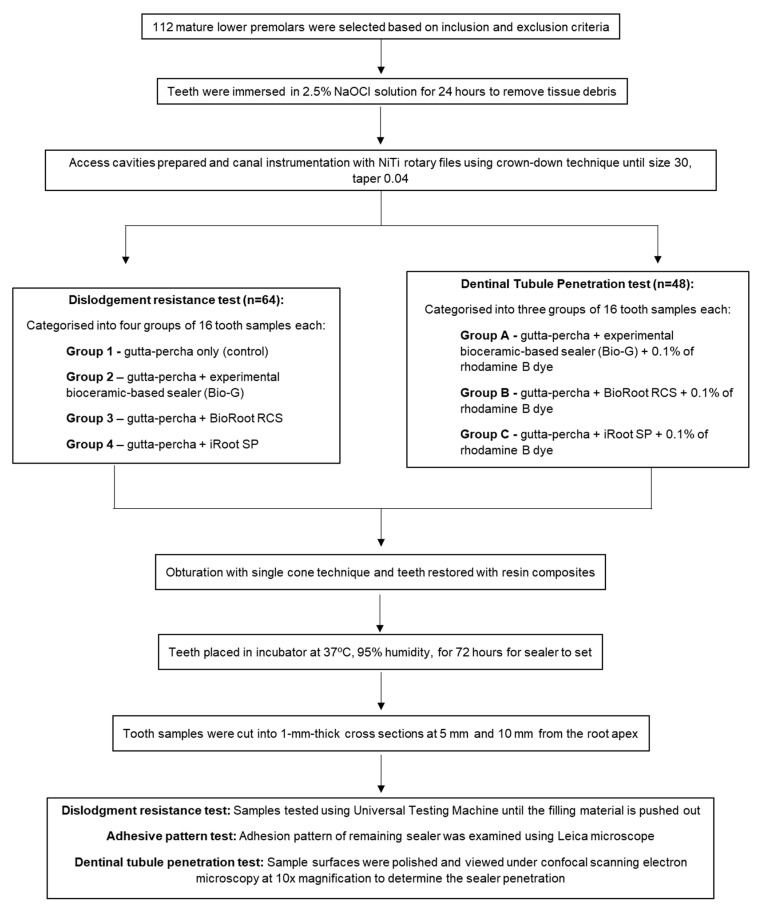
Flowchart of the current study.

**Figure 2 polymers-15-01317-f002:**
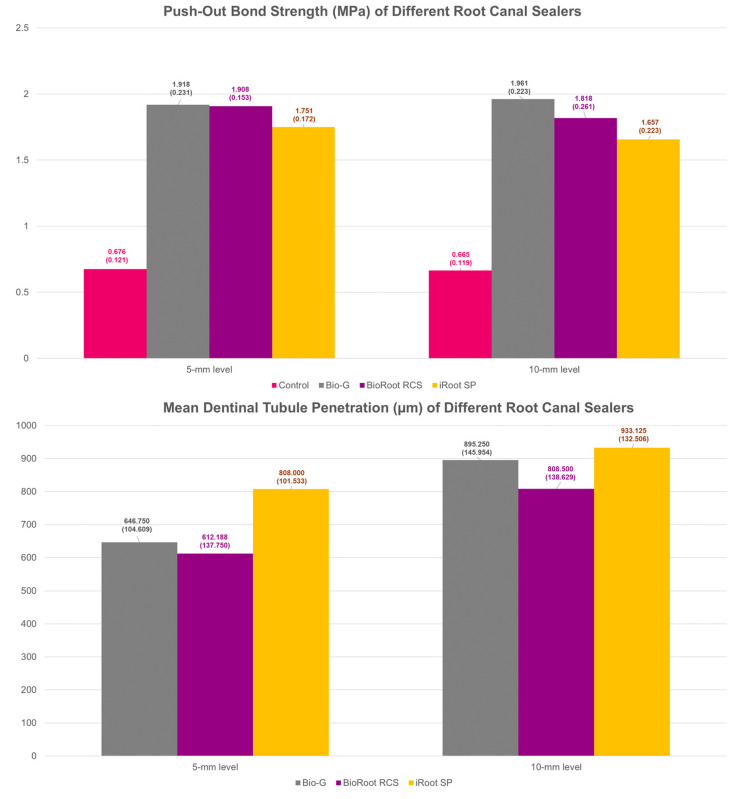
Mean push-out bond strength (MPa) and mean dentinal tubule penetration (µm) of Bio-G, BioRoot RCS, iRoot SP, and the control group.

**Figure 3 polymers-15-01317-f003:**
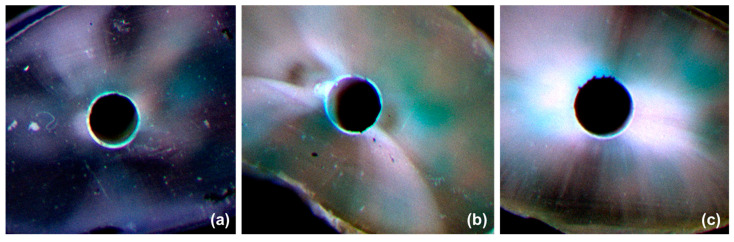
Representative images of the adhesive pattern of Bio-G (**a**), BioRoot RCS (**b**), and iRoot SP (**c**) under Leica microscopy.

**Figure 4 polymers-15-01317-f004:**
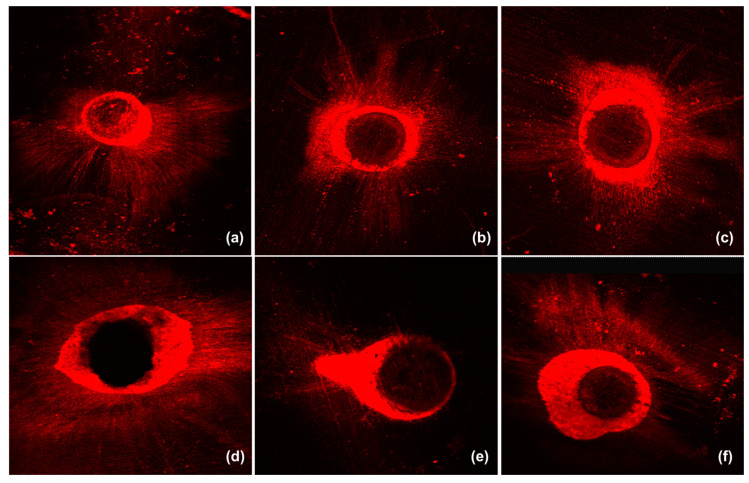
Representative images of sealer penetration under confocal scanning laser microscopy in Bio-G (**a**), BioRoot RCS (**b**), and iRoot SP (**c**) at the 5 mm level and Bio-G (**d**), BioRoot RCS (**e**), and iRoot SP (**f**) at the 10 mm level.

**Table 1 polymers-15-01317-t001:** Mean and standard deviation of push-out bond strength (Nmm^−2^) among different sealers.

Level	Type of Sealer				*p*-Values
	Control	Bio-G	BioRoot RCS	iRoot SP	
5 mm	0.676 ± 0.121	1.918 ± 0.231 ^○^	1.908 ± 0.153 ^○●^	1.751 ± 0.172 ^●^	0.001 *
10 mm	0.665 ± 0.119	1.961 ± 0.223 ^◊^	1.818 ± 0.261 ^◊^ ^♦^	1.657 ± 0.223 ^♦^	0.001 *

* Significant at 0.05; ^○●◊^^♦^ symbols within row indicate no statistical difference (*p* > 0.05).

**Table 2 polymers-15-01317-t002:** Adhesive pattern of Bio-G, BioRoot RCS, and iRoot SP.

Type of Sealer	Type of Adhesive Pattern (*n* = 16)					*p*-Value
	Non-Adhesive	Adhesive				
		1	2	3	4	
**5 mm level**						
Bio-G	-	-	-	10	6	0.001 *
BioRoot RCS	-	-	2	9	5	
iRoot SP	-	2	11	3	-	
**10 mm level**						
Bio-G	-	-	1	9	6	0.001 *
BioRoot RCS	-	-	3	9	4	
iRoot SP	-	4	6	4	2	

* Significant at 0.05.

**Table 3 polymers-15-01317-t003:** Maximum and mean penetration depths (µm) of different sealers.

Level	Bio-G	BioRoot RCS	iRoot SP	*p*-Values
**Maximum Depths**				
5 mm	969.688 ± 78.787	906.375 ± 107.248 ^Ɵ^	1004.500 ± 106.609 ^Ɵ^	0.023 *
10 mm	1439.063 ± 97.380 ^○^	1281.625 ± 187.490	1484.188 ± 141.990 ^○^	0.001 *
**Mean Depths**				
5 mm	646.750 ± 104.609 ^●^	612.188 ± 137.750 ^●^	808.000 ± 101.553	0.001 *
10 mm	895.250 ± 145.954	808.500 ± 138.629 ^╫^	933.125 ± 132.506 ^╫^	0.043 *

* Significant at 0.05; ^Ɵ ╫^ symbols within row indicate statistical difference (*p* < 0.05); ^○●^ symbols within row indicate no statistical difference (*p* > 0.05).

## Data Availability

Not applicable.

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
