# Peer review of "Dislodgment Resistance, Adhesive Pattern, and Dentinal Tubule Penetration of a Novel Experimental Algin Biopolymer-Incorporated Bioceramic-Based Root Canal Sealer"

_polymers, 2023, doi:10.3390/polym15051317_

Round 1

Reviewer 1 Report

Review

Manuscript ID:  polymers-2198055

Journal: polymers

 Dislodgement Resistance, Adhesive Pattern, and Dentinal Tu-2 bule Penetration of a Novel Experimental Algin Biopolymer In-3 corporated Bioceramic-based Root Canal Sealer

The author try to study the dislodgement resistance, adhesive pattern, and dentinal 16 tubule penetration of a novel experimental algin incorporated bioactive-glass 58S calcium silicate-17 based (Bio-G) sealer and compared it with commercialized bioceramic-based sealers. One-hundred-18 and-twelve lower premolars were instrumented to size 30. Four groups (n=16) were assigned for 19 the dislodgment resistance test: control, gutta-percha + Bio-G, gutta-percha + BioRoot RCS, and 20 gutta-percha + iRoot SP, with the exclusion of the control group in adhesive pattern and dentinal 21 tubule penetration tests. Obturation was done and teeth were placed in an incubator to allow sealer 22 setting. For the dentinal tubule penetration test, sealers were mixed with 0.1% of rhodamine B dye. 23 Subsequently, teeth were cut into a 1-mm-thick cross-section at 5-mm and 10-mm levels from the 24 root apex, respectively.

But there are some comments as the following:

1-    The author do not show problem at abstract

2-    The author must add more references to all parts of manuscript

3-    In Keywords there is word hydrogel ,what is the role of hydrogel in manuscript.

4-    There is no knowledge about hydrogel in introduction

5-    The author must show the role of Algin Biopolymer

6-    The author must add the novelty of this work

7-    Figure 3 ,the authors must modify to be clear

8-    The author must modify conclusion .since it is very short.

Recommendation: Major revision

Author Response

1.

The author do not show problem at abstract

The abstract has been amended. However, due to word constraints (200 words max.), the authors could only add a short sentence.

2.

The author must add more references to all parts of manuscript

Several references have been added and the reference number has increased from 38 to …

3.

In Keywords there is word hydrogel, what is the role of hydrogel in manuscript.

The role of alginate hydrogel has been added in the introduction.

Page 2; Line 74-92:

Algin, also known as alginic acid, is a hydrophilic polymer derived from seaweeds that forms a viscous gel-like structure when hydrated [13,14] ……. However, the use of alginate hydrogel in endodontics, particularly for root canal sealers, is still considered new, and there is little evidence available in the literature. Therefore, it is possible to speculate that this com-pact gel-like framework will enable adequate root canal system sealing.”

4.

There is no knowledge about hydrogel in introduction

5.

The author must show the role of Algin Biopolymer

6.

The author must add the novelty of this work

7.

Figure 3, the authors must modify to be clear.

Dear reviewer, we have tried to replace a new image. However, that is the most we can provide. Truly sorry for the inconvenience.

8.

The author must modify conclusion since it is very short.

The conclusion has been modified and two sentences have been added.

Page 12; Line 459-463:

In short, the acceptable dentinal tubules penetration of Bio-G, along with its outstanding dislodgement resistance, may aid to strengthen the seal of the root canal system. However, such a conclusion needs further verification with more well-designed and well-controlled clinical research.”

Reviewer 2 Report

Introduction: it is well written and it offers an overview and state of the art of both the positive and negative aspect of each canal sealer. 

I would only suggest you to add (either in the introduction or discussion) some clinical studies on the topic comparing the different methodologies. The article is currently focusing only on the in vitro application, but most of the techniques have already been tested in clinical investigations.

For this propose I suggest you a recent systematic-review and meta analysis on the topic and I suggest you to add it and cite it to help this aspect.

Mekhdieva, E.; Del Fabbro, M.; Alovisi, M.; Comba, A.; Scotti, N.; Tumedei, M.; Carossa, M.; Berutti, E.; Pasqualini, D. Postoperative Pain following Root Canal Filling with Bioceramic vs. Traditional Filling Techniques: A Systematic Review and Meta-Analysis of Randomized Controlled Trials. J. Clin. Med. 202110, 4509. https://doi.org/10.3390/jcm10194509

Materials and methods: The procedures are well described.

Since the total sample size was divided into smaller subgroups that were assigned to different tests, please add at the beginning of the M&M a flow chart of the study. This can help the readers to immediately understand how the study was designed and what experiments whit how many samples were performed.

Results: well described.

Discussion: the main results and the hypothesis, as well as a description of why the novel Bio-G seals may present some advances compared to the other techniques, are well described. 

I only suggest you to add a short paragraph on the confocal laser scanning analysis including some literature supporting it. Since it is a less known methodology, it can help the readers to deepening the knowledge on the test. Here some reference on the methodology to cite:

https://doi.org/10.3390/app10080761

doi: 10.3390/jcm11061650.

 https://doi.org/10.3390/ma15144850

Author Response

Reviewer 2

9.

Introduction: it is well written, and it offers an overview and state of the art of both the positive and negative aspect of each canal sealer.

I would only suggest you add (either in the introduction or discussion) some clinical studies on the topic comparing the different methodologies. The article is currently focusing only on the in vitro application, but most of the techniques have already been tested in clinical investigations.

For this propose I suggest you a recent systematic-review and meta-analysis on the topic and I suggest you to add it and cite it to help this aspect.

Points have been added to the introduction section.

Page 2; Line 48-53:

Bioceramic-based root canal sealers have gained popularity among clinicians and a systematic review has shown that root canal obturated with bioceramic-based sealer was associated with significantly lower short-term post-operative pain accompanied with lower analgesic intake and flare-up incidence, as compared to root canal-treated teeth obturated with other sealers [6].

10.

Materials and methods: The procedures are well described.

Since the total sample size was divided into smaller subgroups that were assigned to different tests, please add at the beginning of the M&M a flow chart of the study. This can help the readers to immediately understand how the study was designed and what experiments whit how many samples were performed.

A flowchart has been added as Figure 1.

11.

Results: well described.

The authors would like to thank the reviewer for the positive feedback.

12.

Discussion: the main results and the hypothesis, as well as a description of why the novel Bio-G seals may present some advances compared to the other techniques, are well described.

I only suggest you to add a short paragraph on the confocal laser scanning analysis including some literature supporting it. Since it is a less known methodology, it can help the readers to deepening the knowledge on the test. Here some reference on the methodology to cite:

https://doi.org/10.3390/app10080761

doi: 10.3390/jcm11061650.

 https://doi.org/10.3390/ma15144850

A sentence has been added to the discussion section.

Page 11; Line 438-439:

Owing to these advantages, CLSM was used in several in-vitro experimental studies [41-43].”

Round 2

Reviewer 2 Report

Dear Authors,

Thank you for addressing all my comments. I don't have any further concern.